# The Efficacy and Safety of a Microwave Ablation System with a Dipole Antenna Design Featuring Floating Sleeves and Anti-Phase Technology in Stereotactic Percutaneous Liver Tumor Ablation: Results from a Prospective Study

**DOI:** 10.3390/cancers16244211

**Published:** 2024-12-18

**Authors:** Liang Zhang, Lukas Luerken, Vinzenz Mayr, Andrea Goetz, Alexandra Schlitt, Christian Stroszczynski, Ingo Einspieler

**Affiliations:** Department of Radiology, University Medical Center Regensburg, Franz-Josef-Strauss-Allee 11, 93053 Regensburg, Germany; lukas.luerken@ukr.de (L.L.); vinzenz.mayr@ukr.de (V.M.); andrea3.goetz@ukr.de (A.G.); alexandra.schlitt@ukr.de (A.S.); christian.stroszczynski@ukr.de (C.S.)

**Keywords:** liver tumor, hepatocellular carcinoma, liver metastases, microwave ablation, tumor ablation, CT-guided ablation, stereotactic navigation, primary technique efficacy, interventional oncology

## Abstract

Percutaneous microwave ablation (MWA) is a key treatment for primary and secondary liver malignancies. The aim of this prospective study was to evaluate the efficacy and safety of the Surgnova Dophi™ M150E MWA system and to compare ablation defect characteristics with data from the manufacturer’s ablation charts, as well as findings from the literature. In a cohort of 50 patients with 77 liver tumors, stereotactic CT-guided MWA was applied using this novel technology. A primary technique efficacy of 97.4% was achieved, with complete ablation in 75 out of 77 tumors. Complications occurred in 10% of patients, with only 4% classified as major. A comparison with previous in vivo data confirmed the reliability of the system in achieving reproducible and predictable ablation results. In conclusion, the Surgnova Dophi™ M150E system has an excellent efficacy and safety profile for percutaneous stereotactic liver tumor treatment, highlighting its potential in the growing field of interventional oncology.

## 1. Introduction

Percutaneous image-guided thermal ablation has become a cornerstone in the treatment of primary and secondary liver malignancies such as hepatocellular carcinoma (HCC) and colorectal liver metastasis [1,2,3,4]. Recently, the emerging incorporation of navigation systems into percutaneous thermal ablation has begun to represent a significant advancement in interventional oncology, allowing real-time, three-dimensional planning of trajectories and precise targeting of lesions, ensuring accurate and safe probe placement and improvement of the primary technique efficacy (PTE) [5,6,7,8,9,10,11]. The fundamental principle of image-guided thermal ablation is to induce necrosis in all tumor cells while limiting collateral damage to the surrounding liver tissue [12]. Various thermal ablation methods including radiofrequency ablation (RFA) and microwave ablation (MWA), as well as cryoablation, are employed to achieve this goal [13,14,15,16]. Among these, MWA and RFA are the most frequently used techniques, with MWA being increasingly favored due to its advantages over RFA, including faster ablation times, larger ablation areas, and reduced susceptibility to the heat sink effect due to the surrounding blood vessels [13,17,18,19,20,21]. There exist various MWA systems, each with distinct ablation characteristics, including the shape and size of ablation zones, as well as individual safety and efficacy profiles [22,23]. Understanding these characteristics is crucial for maximizing the likelihood of complete tumor ablation, minimizing complications, and optimizing treatment outcomes. However, comprehensive data on these characteristics are lacking for one of the latest state-of-the-art MWA systems, the Surgnova Dophi^TM^ M150E (Surgnova Healthcare Technologies, Beijing, China). This novel system incorporates several advanced technologies. First of all, it features a dipole antenna with floating sleeves that effectively suppresses backward currents, reducing unintended heating along the antenna and ensuring localized energy at the emission point, resulting in a more spherical ablation zone and improved power transfer efficiency [24,25,26,27,28]. Secondly, the patented anti-phase technology helps to create more precise and ultraspherical ablation zones by cancelling out the backward microwave radiation [28]. Thirdly, the system also employs full antenna water cooling, circulating water up to the emission point, which helps to maintain a stable temperature in the ablation zone to improve power delivery, reduce energy loss, and prevent changes in wavelength from tissue damage near the tip [29]. An image and schematic diagram of the antenna are shown in Figure 1. Since only limited data regarding in vivo preclinical, in vivo clinical retrospective, and ex vivo results are currently available for this new MWA technology [28,29,30], this prospective study was conducted to analyze the PTE and safety of the Surgnova Dophi^TM^ M150E MWA system in CT-navigated stereotactic percutaneous liver tumor ablation. Additionally, this study compares the ablation outcomes with data from the manufacturer’s ablation charts, as well as findings from the current literature [28,29,30].

## 2. Methods

### 2.1. Patient Enrollment

Between May 2022 and June 2024, 50 consecutive patients were prospectively enrolled in this single-center study at our institution. The inclusion criteria comprised male patients and non-pregnant, non-lactating female patients aged 18 years or older with primary or secondary liver tumors who were deemed suitable for MWA with curative intention, as reviewed by the interdisciplinary tumor board of the University Hospital of Regensburg. Patients were excluded if they had general contraindications for MWA for liver malignancies, as outlined in the CIRSE standards of practice on thermal ablations for liver malignancies [1]. Additionally, patients with tumors that were ≥5 cm in maximum diameter were also excluded. Written consent was obtained from all participants prior to the procedure. The study was approved by the local ethics committee under approval number 22-2857-101.

### 2.2. MWA Procedure and Follow-Up Imaging

The procedure was performed under general anesthesia in our dedicated interventional CT suite (Somatom Definition Edge, Siemens Healthineers, Forchheim, Germany). Patients were positioned on a vacuum mattress in a supine or slightly right-elevated position. Radiopaque markers were placed on each patient, and sterile draping was applied. A preoperative dual-phase contrast-enhanced planning CT scan, covering the arterial and portal venous phases, was performed after the injection of 100 mL non-ionic contrast medium (Accupaque 350, GE Healthcare Buchler GmbH & Co. KG, Braunschweig, Germany). All CT scans were acquired with the patient in temporary apnea to avoid changes in the liver position due to respiratory movement. The CT data were then transferred to the navigation system (CASOne IR, CAScination AG, Bern, Switzerland) adjacent to the CT gantry to define the ablation trajectories.

Using the CAS-One IR navigation system (CAScination AG, Bern, Switzerland), a Dophi MWA antenna was placed in the target lesion. Before the ablation, an unenhanced CT scan was performed to verify the correct position of the antenna, with manual repositioning executed as necessary. The ablation parameters were tailored to achieve a minimum ablative margin of 5 mm [1], based on the manufacturer’s ablation charts (i.e., ex vivo preclinical data from bovine liver), which are integrated into the CAS-One system. After the ablation, a track ablation was performed to coagulate the needle tract and minimize the risk of tumor seeding along the antenna path. An immediate post-procedural CT scan, conducted in the same manner as the preoperative planning CT scan, was performed after antenna removal to rule out any peri-interventional complications and to confirm technical success.

Follow-up imaging was first performed 6 weeks post-procedure, using multiphase MRI with gadoxetic acid (Primovist, Bayer AG, Leverkusen, Germany) as the contrasting agent. The imaging protocol included the arterial (10 s), late arterial (40 s), portal venous (75 s), and hepatobiliary late (20 min) phases. In rare cases, a CT scan was performed due to contraindications for MRI. Figure 2 illustrates a representative case of microwave ablation treatment.

### 2.3. Endpoints and Statistical Analysis

The primary endpoint was the PTE [31], which was defined as the complete eradication of the target lesion as observed in the initial follow-up imaging session 6 weeks after treatment. Secondary endpoints were the number of complications, classified and graded according to the Cardiovascular and Interventional Radiological Society of Europe’s classification system (CIRSE) [32], and the number of procedure-related side effects (i.e., minimal asymptomatic perihepatic hematoma, asymptomatic pleural effusion, pain, and post-ablation syndrome) [31]. In addition, the size and shape of the ablation defects depending on different ablation settings were evaluated in relation to the manufacturer’s ablation charts and results from the current literature. The two-dimensional ablation defect size and the ablation defect sphericity index (SI) were obtained from multiplanar post-procedural CT images by measuring the long axis (i.e., longest diameter) and short axis (i.e., shortest diameter). The SI was calculated by dividing the short axis by the perpendicular long axis, as described previously [28]. The ablation defect volume was manually determined using freehand volumetry in Syngo via (Siemens Healthineers, Forchheim, Germany).

Normally distributed continuous data are presented as the mean ± standard deviation. Pearson’s correlation coefficient was used to evaluate linear relationships between ablation parameters (i.e., ablation power, ablation duration, and energy dose, defined as the product of ablation power and ablation duration) and ablation defect characteristics. Statistical analysis was performed using IBM SPSS 28.0 (IBM Corp, Armonk, NY, USA). A *p*-value of <0.05 was considered statistically significant.

## 3. Results

### 3.1. Patient Demographics, Tumor and Treatment Characteristics, and Lesion Distribution

The patient demographics, tumor characteristics, and key procedural metrics are summarized in Table 1, while tumor distribution is illustrated in Figure 3. Among the tumors, 57.1% were situated near the liver capsule (within 10 mm), 8.1% were subdiaphragmatic or subcardial (within 10 mm), and 22.1% were adjacent (within 5 mm) to major blood vessels (>3 mm in diameter). Four tumors were adjacent to other organs (within 10 mm), including two near the stomach, one near the gallbladder, and one near the kidney. Of the seventeen tumors adjacent (<5 mm) to vessels (>3 mm in diameter), eight were near a portal vein branch, six were near a hepatic vein, two were near the vena cava, and one was near a hepatic artery branch.

### 3.2. Primary Technique Efficacy

Complete tumor eradication at first follow-up was achieved in 75 of the 77 tumors, resulting in a PTE of 97.4%. In the two cases (both subcapsular colorectal liver metastases) of incomplete ablation, local tumor progression (LTP) was observed at the edge of the ablation zone during the 6-week follow-up MRI, suggesting an insufficient minimal ablative margin, which was not evident at the time of the first post-procedural CT scan due to subcapsular liver deformation [31].

### 3.3. Complications and Side Effects

A total of five complications occurred in five patients. Minor CIRSE grade 1 complications occurred in three patients, and major CIRSE grade 3 complications occurred in two patients. No grade 2, 4, 5, or 6 complications were observed. A detailed overview of the complications is provided in Table 2.

A total of 40 side effects were observed in 34 patients. Of these, 31 patients (62%) experienced mild to moderate temporary pain; 7 patients (14%) had minor perihepatic hematoma; and 2 patients (4%) had asymptomatic pleural effusion. No delayed complications or side effects (>30 days after ablation) were reported.

### 3.4. Ablation Defect Size, Volume, and SI

The mean dimensions of the ablation defects were as follows: the long axis measured 40.2 ± 7.8 mm, the short axis measured 30.9 ± 6.5 mm, and the volume was 26.3 ± 13.0 cm^3^. The mean SI was 0.77 ± 0.11. In 86% (66/77) of the tumor ablations, the SI was greater than 0.66. The ablation defect dimensions and SI, stratified by ablation parameters, are detailed in Table 3.

Detailed comparisons between the ablation defect characteristics of the current study (“C”) and those of the manufacturer’s ablation charts (i.e., ex vivo bovine liver data); the previously published in vivo porcupine liver data from Habert et al. [29]; the human clinical data from Blain et al. [28]; and the ex vivo bovine liver data from Namakshenas et al. [30], along with the deviation (Dev) between the studies, are presented in Table 4, Table 5, Table 6 and Table 7. Ablation defect volume and SI are compared only when such data are available and sufficiently comparable across the studies.

Ablation defect volume correlated significantly with ablation power (r = 0.46, *p* < 0.001), ablation duration (r = 0.62, *p* < 0.001), and energy dose (r =. 0.64, *p* < 0.001). There were no significant correlations between SI and ablation power (r = −0.05, *p* = 0.70), ablation duration (r = −0.1, *p* = 0.40), or energy dose (r = −0.07, *p* = 0.56).

## 4. Discussion

This study represents the first prospective evaluation of the Surgnova Dophi™ M150E MWA system, providing a crucial contribution to the limited body of literature on this novel technology. A PTE of 97.4% and a major complication rate of 4% demonstrate its outstanding effectiveness and safety profile in stereotactic percutaneous liver tumor ablation, highlighting its potential in the growing field of interventional oncology.

Unlike previous studies, which have primarily relied on in vivo preclinical, in vivo clinical retrospective, and ex vivo data [28,29,30], this is a prospective single-center study using CT-guided stereotactic navigation in the same CT suite, ensuring standardized treatment conditions and minimizing variability in both technique and outcomes. The PTE of 97.4% observed in this study is comparable or superior to that reported in studies that have used various targeting and imaging methods, including stereotactic and free-hand approaches under image guidance by CT, MRI, and ultrasound [6,11,33,34,35,36,37]. Notably, a recent meta-analysis by Tinguely et al. demonstrated a pooled PTE of 94% when assessed at 1–6 weeks and 90% when assessed at 6–12 weeks following the stereotactic and robotic minimally invasive ablation of malignant liver tumors, which is in line with our results [6].

The overall complication rate of 10% (5/50) in this study is concordant with the pooled estimate of 11.4% from previously published data on stereotactic and robotic liver ablation [6]. Minor complications (i.e., CIRSE grade 1) occurred in 6% (3/50) of patients. Of note is the fact that two of these patients required interventional treatment to resolve the complication. Since both complications could be solved within the same session without needing subsequent additional therapy or causing a deviation from the normal post-therapeutic course, they were classified as CIRSE grade 1. Major complications (i.e., CIRSE grade 3), affecting 4% (2/50) of patients, necessitated further intervention after the ablation session. Specific cases included a liver abscess and one case of liver hemorrhage, which could be treated sufficiently without post-procedure sequelae. This complication rate is consistent with the range of experiences reported for different thermal ablative systems [6,38], indicating that the Surgnova Dophi™ M150E’s safety profile aligns with current standards in the field. For the sake of completeness and clarity, ablation-related side effects were reported in a total of 34 patients in our study. The number of side effects is in line with those reported in the literature. A study by Liang et al. observed local pain in 80.1% of patients and asymptomatic pleural effusion in 10.4% after ablation within a cohort of 1136 patients, compared to 62% and 4%, respectively, in our study [39]. According to Ahmed et al. [31], side effects—undesired consequences of the procedure that occur frequently but rarely, if ever, result in significant morbidity—are expected. We believe it is crucial to emphasize the difference between side effects and true complications, as only the latter usually have a relevant impact on further management. To date, the use of different definitions and terminology for safety and complications [31,32,40] severely limits the comparability between studies, highlighting the critical need for further standardization of adverse event definitions, as noted by Tinguely et al. [6].

The long axis of the ablation defect measured in this study was consistent with the values based on ex vivo bovine liver data reported in the manufacturer’s ablation charts and by Namakshenas et al. [30]. However, the short axis of the ablation defect was consistently smaller, resulting in a lower SI and ablation defect volume compared to the ideal spherical ablation model depicted in the Surgnova ablation charts and the near-spherical ablation shapes of Namakshenas et al. [30]. This discrepancy between ex vivo data and our in vivo data is not uncommon and aligns with broader observations in the field [41]. For instance, a study by Amabile et al. on the HS Amica ablation system demonstrated a significant reduction in SI when moving from ex vivo to in vivo experiments [29,42]. Such differences are likely due to the complex and variable nature of in vivo conditions, where factors like tissue heterogeneity and heat sink effects from blood flow influence the final ablation shape [41]. By contrast, the ablation defect sizes observed in this study correspond closely with in vivo data from both Habert et al. and Blain et al., highlighting the consistency of the Surgnova Dophi™ M150E MWA system in achieving reproducible and predictable ablation results [28,29]. For example, at an ablation setting of 100 Watts for 10 min, the long axis differed by only 0.9 mm from Habert’s and 1.2 mm from Blain’s data. Similarly, deviations in the short axis were small, with a 3.4 mm difference from Habert’s results and a 2.4 mm difference from Blain’s [28,29]. It is noteworthy that the ablation defect volumes reported in this study were larger than those reported by Habert et al. [29]. This discrepancy is likely due to differences in the methodology used to measure the ablation volume. In our study, we employed freehand volumetry, which allows for more detailed and precise measurement of the actual ablation zone. In contrast, Habert et al. used the ellipsoid volume formula, which assumes a more regular, idealized shape for the ablation defect. This may have resulted in an underestimation of the actual ablation volume [29].

As a major observation, the mean SI was 0.77 ± 0.11, indicating that the ablation defects were relatively well-rounded, though not perfectly spherical. Notably, 86% (66/76) of the ablation defects demonstrated an SI greater than 0.66. These results are closely aligned with the SI reported by previous in vivo studies: Blain et al. reported a mean SI of 0.78 ± 0.14, with 82% of the ablation defects having an SI greater than 0.66 [28]. Additionally, when using the same calculation method as Habert et al. [29], which defines SI as the square of the short axis diameter divided by the square of the long axis diameter, the median SI in our study was 0.58, precisely matching the median SI reported by Habert et al. [29] Of note is the fact that our results do not show a significant correlation between SI and ablation power, ablation duration, or energy dose. This is likely due to the antenna design features described above, which result in relatively spherical defects regardless of the chosen ablation parameters.

This study has some limitations. First, as a single-center study, the findings may not be fully generalizable to other clinical environments with different equipment, operator experience, or patient populations. The relatively small sample size may also limit the ability to detect rare complications or variations in treatment efficacy. Additionally, the study focused on short-term outcomes, and longer follow-up is needed to assess long-term efficacy, including local tumor progression and overall survival rates. Moreover, the study lacks a comparison group including other microwave ablation systems, limiting insights into its relative efficacy.

## 5. Conclusions

This prospective study demonstrates that stereotactic percutaneous MWA with the Surgnova Dophi™ M150E system is both safe and effective for the treatment of liver tumors. The PTE of 97.4% is consistent with, and in some cases superior to, previously published data on stereotactic liver ablations. Furthermore, the system’s complication rates are within the expected range for MWA procedures, underlining its safety profile. The ablation defect sizes and SI observed in this study closely match those from prior in vivo research, confirming the system’s reliability in achieving reproducible and predictable ablation outcomes. The discrepancies between our data and the ex vivo data, particularly in terms of the ablation defect’s short axis, SI, and volume, are likely due to the inherent complexities of in vivo conditions, such as tissue heterogeneity and perfusion resulting in the heat sink effect, which are absent in ex vivo environments. Overall, the Surgnova Dophi™ M150E MWA system is a robust and reliable option for liver tumor ablation, with performance metrics comparable to established benchmarks in the field.

## Figures and Tables

**Figure 1 cancers-16-04211-f001:**
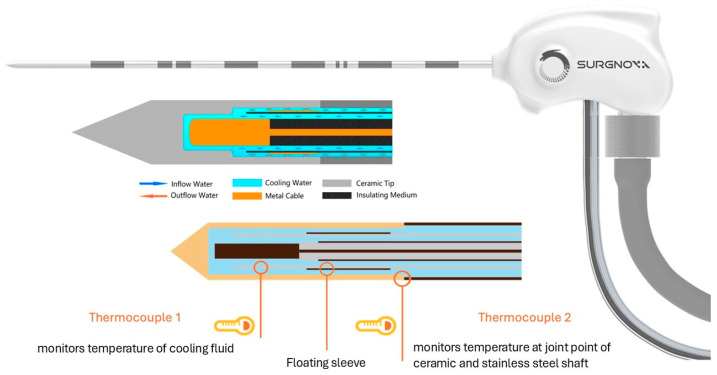
Image and schematic diagram of the Dophi ablation antenna. The figure presents an image of the Dophi ablation antenna and a schematic diagram of its tip. The schematic highlights key components such as the dipole antenna with its floating sleeve, the water cooling system, and the two thermocouples that monitor the cooling fluid’s temperature and the joint temperature between the ceramic tip and the stainless steel shaft. Images are adapted from vivamus-medical GmbH (Bensheim, Germany).

**Figure 2 cancers-16-04211-f002:**
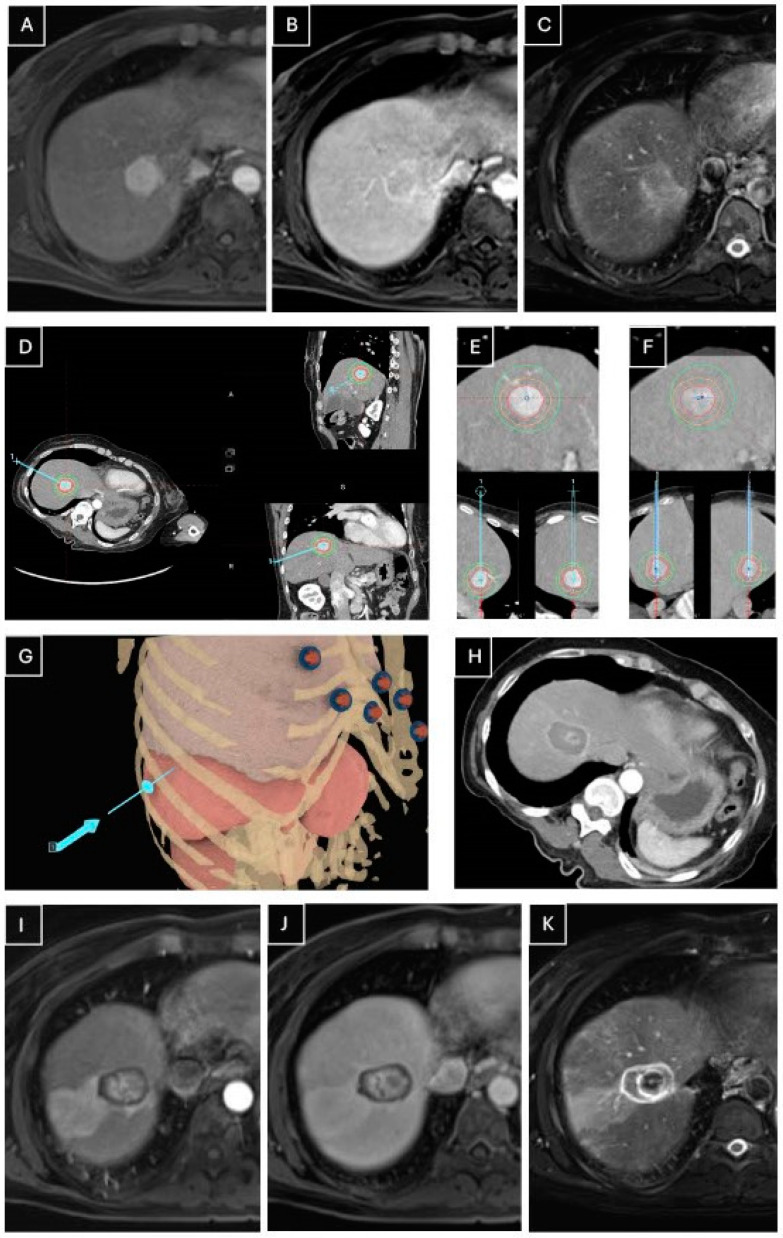
Stereotactic microwave ablation of a 60-year-old female with a 22 mm hepatocellular carcinoma (HCC) in segment VIII. A preoperative MRI scan shows the lesion in (**A**) the arterial phase, (**B**) the portal venous phase, and (**C**) T2-weighted imaging, with arterial hypervascularization, washout in the portal venous phase, and intermediate signal intensity on T2. (**D**,**E**): multiplanar reconstructions and radial reconstructions along the axis of the planned trajectory of the preoperative planning CT in the arterial phase, outlining the planned trajectory. (**F**): non-contrast-enhanced CT after placement of the antenna, fused with the preoperative CT scan, confirming the adequate placement of the antenna. (**G**) A 3D model showing the microwave antenna’s trajectory. (**H**) Immediate post-procedural CT in the arterial phase confirming complete ablation. Follow-up MRI scans at 2 months show the arterial phase (**I**), the portal venous phase (**J**), and T2-weighted imaging (**K**) with no recurrence.

**Figure 3 cancers-16-04211-f003:**
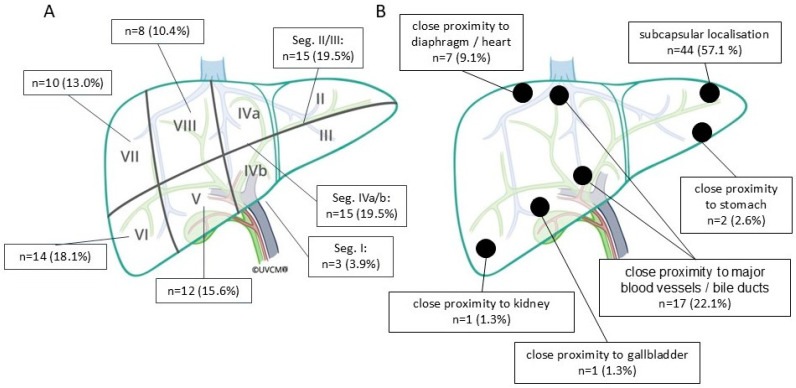
(**A**) The distribution of treated tumors within the liver, categorized by liver segments. (**B**) The distribution of treated tumors based on their proximity to adjacent anatomic structures. The figure was adapted with permission from Lachenmayer et al. and Inselspital, Bern University Hospital [33].

**Table 1 cancers-16-04211-t001:** An overview of the patient demographics, tumor characteristics, and key procedural parameters. Abbreviations: SD—standard deviation, HCC—hepatocellular carcinoma, CCC—cholangiocarcinoma, DLP—dose–length product.

Characteristics	Value
Age, mean ± SD (range)	64.9 ± 10.2 years (43–87 years)
**Gender**	
Female	13
Male	37
Number of tumors treated (n = number of patients)	77 (n = 50)
**Tumor entity**	
**Primary**	52 (n = 38)
HCC	51 (n = 37)
CCC	1 (n = 1)
**Metastatic**	25 (n = 12)
Colorectal	19 (n = 9)
Leiomyosarcoma	3 (n = 1)
Angiosarcoma	2 (n = 1)
Neuroendocrine tumor	1 (n = 1)
Tumor long axis, mean ± SD (range)	17.0 ± 7.4 mm (3–33 mm)
Tumor short axis, mean ± SD (range)	12.9 ± 6.0 mm (2–31 mm)
Ablation power, mean ± SD (range)	92.2 ± 14.2 Watts (50–100 Watts)
Ablation time, mean ± SD (range)	7.7± 2.2 min (1.5–10 min)
No. of control scans per tumor, mean ± SD (range)	2.6 ± 1.8 scans (1–11 scans)
DLP, mean ± SD (range)	2006 ± 620 mGy·cm (952–3572 mGy·cm)
Intervention time, mean ± SD (range)	89 ± 38 min (44–240 min)
Hospital stay after intervention, mean ± SD (range)	2.0 ± 0.77 days (1–4 days)

**Table 2 cancers-16-04211-t002:** A summary of the complications associated with the ablation procedure, categorized by CIRSE complication grade and onset time, along with descriptions of each complication. The onset time was classified as immediate (occurring 6–24 h after ablation), periprocedural (within 30 days following ablation), or delayed (>30 days after ablation) [31].

CIRSE Complication Grade	Complication Onset	Description
1	Immediate	Minor abdominal wall bleeding several hours after the ablation, requiring no further treatment.
1	Immediate	Active arterial bleeding at the site of ablation, treated with transcatheter arterial embolization using a combination of Glubran^®^ 2 (GEM Srl, Viareggio, Italy) and Lipiodol, resolved within same session as the ablation.
1	Immediate	Pneumothorax during the ablation, treated with a chest tube, resolved within the same session as the ablation.
3	Immediate	Active arterial bleeding several hours after the ablation, treated with transcatheter arterial embolization using coils.
3	Periprocedural	Liver abscess one week after the ablation, effectively treated with antibiotics and percutaneous CT-guided abscess drainage.

**Table 3 cancers-16-04211-t003:** The ablation defect long axis, short axis, volume, and SI, as well as number of tumors treated, stratified by ablation parameters.

Ablation Parameters	Ablation Defect Long Axis (mm)		Ablation Defect Short Axis (mm)		Ablation Defect Volume (cm^3^)		Ablation Defect SI		Number of Tumors
	Mean	SD	Mean	SD	Mean	SD	Mean	SD	
100 W 10 min	45.9	5.3	35.4	4.8	34.2	9.4	0.77	0.09	29
100 W 8 min	40.5	4.6	30.6	4.6	30.5	14.8	0.77	0.14	14
100 W 7 min	33.0	1.4	24.0	9.9	12.1	1.5	0.72	0.27	2
100 W 5 min	36.1	7.3	27.5	4.9	19.5	6.4	0.77	0.09	12
75 W 10 min	44.0	-	33.0	-	35.0	-	0.75	-	1
75 W 8 min	38.3	7.4	33.7	9.3	34.6	14.5	0.87	0.12	3
75 W 7 min	42.5	3.5	27.5	2.1	19.8	1.2	0.65	0.10	2
75 W 5 min	34.7	6.3	27.9	4.6	14.6	4.9	0.81	0.08	10
50 W 10 min	32.0	-	18.0	-	13.1	-	0.56	-	1
50 W 5 min	26.5	9.2	20.0	1.4	4.3	0.1	0.79	0.22	2
50 W 1.5 min	22.0	-	19.0	-	3.6	-	0.86	-	1

**Table 4 cancers-16-04211-t004:** A comparison between the ablation defect dimensions (long axis, short axis) and volume of the current study (“C”) and the manufacturer’s charts (“Surgnova”) based on ex vivo bovine liver data with matching ablation power and duration settings. The volume of the Surgnova ablation zone was calculated using the long axis diameter, as reported in the manufacturer’s charts, assuming that the ablation zone forms a perfect sphere. This was done by applying the formula for the volume of a sphere: 4/3 π (long axis diameter/2)^3^. Deviations (Dev) indicate differences between the two datasets.

Parameters	Long Axis (mm)			Short Axis (mm)			Volume (cm^3^)		
	C	Surgnova	Dev	C	Surgnova	Dev	C	Surgnova	Dev
100 W 10 min	45.9	45	0.9	35.4	45	−9.6	34.2	47.7	−13.5
100 W 8 min	40.5	40	0.5	30.6	40	−9.4	30.5	33.5	−3
100 W 5 min	36.1	35	1.1	27.5	35	−7.5	19.5	22.4	−2.9
75 W 10 min	44.0	40	4.0	33.0	40	−7.0	35.0	33.5	1.5
75 W 8 min	38.3	35	3.3	33.7	35	−1.3	34.6	22.4	12.2
75 W 5 min	34.7	30	4.7	27.9	30	−2.1	14.6	14.1	0.5
50 W 10 min	32.0	35	−3.0	18.0	35	−17	13.1	22.4	−9.3
50 W 5 min	26.5	25	1.5	20.0	25	−5	4.3	8.2	−3.9

**Table 5 cancers-16-04211-t005:** A comparison between the ablation defect dimensions (long axis, short axis) and volume of the current study (“C”) and the data from Habert’s in vivo swine model for microwave ablation in liver tissue (“Habert”) with matching ablation power settings [29]. Deviations (Dev) indicate differences between the two studies’ measurements.

Parameters	Long Axis (mm)			Short Axis (mm)			Volume (cm^3^)		
	C	Habert	Dev	C	Habert	Dev	C	Habert	Dev
100 W 10 min	45.9	45	0.9	35.4	32	3.4	34.2	24.3	9.9
100 W 8 min	40.5	38	2.5	30.6	29	1.6	30.5	16.3	14.2
75 W 10 min	44	38	6	33	27	6	35	14.9	20.1
50 W 10 min	32	31	1	18	25	−7	13.1	10.1	3
50 W 5 min	26.5	29	−2.5	20.0	22	−2	4.3	7.3	−3

**Table 6 cancers-16-04211-t006:** A comparison between the ablation defect dimensions (long axis, short axis) of the current study (“C”) and the data from Blain’s in vivo retrospective study (“Blain”) with matching ablation power settings [28]. Deviations (Dev) indicate differences between the two studies’ measurements.

Parameters	Long Axis (mm)			Short Axis (mm)		
	C	Blain	Dev	C	Blain	Dev
100 W 10 min	45.9	44.7	1.2	35.4	33	2.4
100 W 8 min	40.5	39.5	1.0	30.5	28.8	1.7
75 W 10 min	44	37.1	6.9	33	27.5	5.5
50 W 5 min	26.5	24.3	2.2	20.0	18.9	1.1

**Table 7 cancers-16-04211-t007:** A comparison between the ablation defect dimensions (long axis, short axis) and sphericity index (SI) of the current study (“C”) and the ex vivo data reported by Namakshenas et al. (“Namak-shenas”) with matching ablation power settings and durations [30]. The SI in Namakshenas’ study was retrospectively calculated by dividing the shorter diameter by the longer diameter [30]. Deviations (Dev) indicate differences between the two studies’ measurements.

Parameters	Long Axis (mm)			Short Axis (mm)			SI		
	C	Namakshenas	Dev	C	Namakshenas	Dev	C	Namakshenas	Dev
100 W 10 min	45.9	42	3.9	35.4	41	−5.6	0.77	0.98	−0.21
100 W 5 min	36.1	36	0.1	27.5	33	−5.5	0.77	0.92	−0.15
50 W 10 min	32.0	33	−1	18.0	33	−15	0.56	1	−0.44
50 W 5 min	26.5	29	−2.5	20.0	25	−5	0.79	0.86	−0.07

## Data Availability

The data presented in this study are available on request from the corresponding author. The data are not publicly available due to data protection requirements and legal constraints.

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
