# Peer review of "The Efficacy and Safety of a Microwave Ablation System with a Dipole Antenna Design Featuring Floating Sleeves and Anti-Phase Technology in Stereotactic Percutaneous Liver Tumor Ablation: Results from a Prospective Study"

_cancers, 2024, doi:10.3390/cancers16244211_

Round 1

Reviewer 1 Report

Comments and Suggestions for Authors

Thank you very much for the opportunity to review the paper “Efficacy and Safety of a Microwave Ablation System with Dipole Antenna Design Featuring Floating Sleeves and Anti-Phase Technology in Stereotactic Percutaneous Liver Tumor Ablation: Results from a Prospective Study”

Overall, the paper is well-written, and the conclusions are sound. I have a few minor suggestions to improve the manuscript further.

1.      If possible, the addition of a schematic diagram of the ablation antenna (lines 57-64) in the introduction would be helpful to the readers.

2.      Please add an annotation to Table 1 that explains the abbreviations used in the table such as DLP.

Reviewer 2 Report

Comments and Suggestions for Authors

Nice paper.

Complication rate seems high. Can you please make a table of complications and discuss and compare with the literature available.

Can you add a picture of the microwave needle.

Can you add a CT picture of the liver with the needle.

Can you add a paragragh about different microwave systems available with pros and cons of each system.

Reviewer 3 Report

Comments and Suggestions for Authors

The authors present the outcomes from a single center prospective study focusing on the safety and efficacy of a specific MWA system in CT-navigated stereotactic percutaneous liver tumor ablation. This is an interesting, well-designed and somewhat well-written study. A number of issues to be addressed:

- I think the manuscript would benefit from a figure with lesion locations within the liver

- The authors should also clarify vicinity with major vessels specifically in their ablated lesions

- The manuscript could also benefit from some figures of the ablated lesions pre and post mw to show the evolution of their process

- There should be a follow up section within the results to clarify if there were any late adverse events or not

- There is no information about the hospital stay

- The limitations section needs to be improved ie there is no comparison group with another mwa system etc
